# Tomato Natural Resistance Genes in Controlling the Root-Knot Nematode

**DOI:** 10.3390/genes10110925

**Published:** 2019-11-14

**Authors:** Ahmed H. El-Sappah, Islam M. M., Hamada H. El-awady, Shi Yan, Shiming Qi, Jingyi Liu, Guo-ting Cheng, Yan Liang

**Affiliations:** 1College of Horticulture, Northwest A&F University, Yangling 712100, China; Ahmed_elsappah2006@yahoo.com (A.H.E.-S.); monirul@nwafu.edu.cn (I.M.M.); hmada93elawdy@gmail.com (H.H.E.-a.); qishiming2008@163.com (S.Q.); liujingyi1987@nwsuaf.edu.cn (J.L.); chengguoting@163.com (G.-t.C.); 2State Key Laboratory of Crop Stress Biology in Arid Regions, Northwest A&F University, Yangling 712100, China; 3Genetics Department, Faculty of Agriculture, Zagazig University, 44511 Zagazig, Egypt

**Keywords:** root-knot nematode, tomato-resistant resources, *Mi* resistance genes, heat-stable resistance, gene-based marker

## Abstract

The root-knot nematode (RKN) is one of the most dangerous and widespread types of nematodes affecting tomatoes. There are few methods for controlling nematodes in tomatoes. Nature resistance genes (R-genes) are important in conferring resistance against nematodes. These genes that confer resistance to the RKN have already been identified as *Mi-1*, *Mi-2*, *Mi-3*, *Mi-4*, *Mi-5*, *Mi-6*, *Mi-7*, *Mi-8*, *Mi-9*, and *Mi-HT*. Only five of these genes have been mapped. The major problem is that their resistance breaks down at high temperatures. Some of these genes still work at high temperatures. In this paper, the mechanism and characteristics of these natural resistance genes are summarized. Other difficulties in using these genes in the resistance and how to improve them are also mentioned.

## 1. Introduction

Tomato is part of the Solanaceae family and the second most common vegetable after potatoes in global food production. Tomatoes are a rich source of micronutrients, such as minerals, vitamins, and antioxidants that are essential for the human diet. They also contain high levels of lycopene, an antioxidant that reduces the risks associated with many cancers and neurological diseases [1]. Many pests and diseases affect both the quantity and quality of tomato production. Plant-parasitic nematodes are one of them.

The root-knot nematode (RKN) belongs to the genus *Meloidogyne*, which includes more than 90 species, and some of them have several races. The four species *M. incognita*, *M. javanica*, *M. hapla*, and *M. arenaria* are considered the most economically devastating worldwide. They are biotrophic parasites capable of infecting more than 2000 plant species. *Meloidogyne* pp. was reported for the first time in Cassava [2]. In most crops, nematode damage affects the plant health and growth. The parasites infect plant roots and induce the formation of giant feeding cells leading to a decrease in plant nutrition and water uptake. As a consequence, plants can show several symptoms such as wilting and stunting, thus increasing the susceptibility to other pathogens and with considerably reduced yields.

RKN, like most other species of plant-parasitic nematodes, has a relatively simple life cycle consisting of the egg, four juvenile stages, and the adult male and female. Near or at the surface of the root, the RKN’s females deposit eggs in a gelatinous mass. The juvenile hatches and migrates either to a different location in the root or to the soil. Every juvenile penetrates a suitable root by repeatedly thrusting its stylet into the cells of the root-surface. Within a few days, the juvenile becomes settled with its head embedded in the developing vascular system, and it begins feeding [3]. As a result of this, the cell response, by secreting different enzymes, causes an increase in cell size and number (giant cells). As the nematode matures, the male reverts to worm-shaped, and the females begin laying eggs. Meanwhile, RKNs slowly move through the soil, and the nematode changes its location in the soil either by its movements as a living organism or by external factors (equipment, shoes or boots, etc.) [3].

RKNs can cause severe damage to a plant, especially to the roots. Tomato varieties have different responses toward various *Meloidogynes* pp. Symptoms are more prevalent with tropical species compared to those in the temperate or cold regions. Damage and yield loss studies conducted for have shown significant differences in the degree of susceptibility among tomato cultivars. Moreover, different populations of the same species of *Meloidogyne* even exhibit different degrees of pathogenicity on a specific tomato cultivar [4].

Resistance in tomatoes to RKNs was first observed by Bailey [5] in the wild species (*Lycopersicon peruvianum* L.) Mill. P.I. 128657. Using embryo rescue, Smith [6] introduced this trait into the domesticated *Lycopersicon esculentum*. This resistance was subsequently shown to been coded by a single dominant gene called *Meloidogyne incognita*-1 (*Mi-1*) located in chromosome 6 [7], which has important activities against *M. incognita*, *M. arenaria*, and *M. javanica*, but not *M. hapla* [8]. *Mi-1* is currently the only commercially available source of resistance. The resistance provided by the *Mi* gene is induced at an early stage in tomato plants as early as two weeks after germination. This gene detected resistance in the leaves and roots of tomato plants [9]. Other resistance genes (*Mi-2*, *Mi-3*, *Mi-4*, *Mi-5*, *Mi-6*, *Mi-7*, *Mi-8*, *Mi-HT*, and *Mi-9*) have also been identified. There is a suggestion that wild tomato has a genetic system with an ability to generate variation at the nematode resistance locus, leading to the generation of new resistance specifications [10]. Of the ten genes for resistance to RKNs identified in tomatoes, seven (*Mi-2*, *Mi-3*, *Mi-4*, *Mi-5*, *Mi-6*, *Mi-9*, and *MI-HT*) show heat-stable resistance [11]. *Mi-3* is mapped to the short arm of chromosome 12 [12]. The same of *Mi-3* and *Mi-5* are located on chromosome 12. *Mi-9* is a homolog of *Mi-1* [13]. It has been mapped to the short arm of chromosome 6 between markers C32.1 and C&B [14]. *Mi-HT* is mapped to the short arm of chromosome 6. The other three heat-stable resistance genes have not been mapped, and none of these genes has been cloned to date. In this article, we mention all these genes concerned with RKN resistance in tomato.

## 2. Distribution of *Meloidogyne* Species around the World

The root-knot nematode is a microscopic parasitic species of *Meloidogyne*. *Meloidogyne* species are widely distributed around the world (Figure 1) [4,15,16]. They are dangerous for two reasons: first, there are a large number of species (about 98), which leads to an abundance in most climates and many countries. The second one is that they have many plant hosts—more than 2000 plant species [16]. Agriculture is the most widespread method of spreading the pests, which include the movement of tubers, transplants, and soil, and in fresh and sea water [16,17]. They are mostly distributed in the tropical countries [4]. They are more abundant within the first 1 m of soil [16]. The most common species are *Meloidogyne javanica*, *M. incognita*, *M. hapla*, and *M. arenaria*. In tropical countries, the most distributed species are *M. incognita* and *M. javanica*; in cold countries, with a temperature of less than 15 °C, *M. hapla* is the most distributed [18]. *Meloidogyne* species are pathogens of economic importance, ranked as the most dangerous plant parasitic nematode worldwide, listed as the number one in a survey carried out by Jones et al. [19] followed by cyst nematodes (*Globodera* and *Heterodera* spp.), root-lesion nematodes (*Pratylenchus* spp.), the reniform nematode *Rotylenchulu sreniformis*, *Nacobbus aberrans*, the pine wilt nematode *Bursaphelenchus xylophilus*, the burrowing nematode *Radopholus similis*, *Xiphinema index*, *Ditylenchus dipsaci*, and *Aphelenchoide besseyi* [19].

## 3. Naturally Resistant Resources

Several *Mi*-genes have been detected in some tomato lines, genotypes, and cultivars. These genes confer resistance against root-knot nematodes (Table 1). Many resources of resistance have been discovered from the first one (*L. peruvianum*) PI128657) since 1944. Which resistance genes some of these plants contain is still not known. Some of the plants confer resistance at high temperatures, such as (*L. peruvianum*) PI126443, ZN48, and LA0385. The preferred and safest method for controlling RKNs is in the discovery of new resistant plants. It is important to perform an extensive evaluation of tomato plants whose resistance has not been determined.

## 4. The Mechanism of Natural Resistance

Tomatoes, like all plants, undergo several modes for protection and immunity (Figure 2). The plant has an innate immune system that can recognize pathogen-associated molecular patterns (PAMPs) [29]. PAMP-triggered immunity (PTI) is the first defense line of response of the plant to pathogens. The extra cellular receptor proteins, receptor-like kinases (RLK), and receptor-like protein (RLP) are initiation factors and activators of the first defense line [30]. The second defense line is triggered by intracellular proteins that contain a nucleotide-binding site (NBS), a toll-like interleukin receptor (TIR), which is not found in the *mi-1* gene, and leucine-rich repeats (LRRs) [30]. During the second-line defense, there are two modes of pathogen interaction: direct and indirect [31]. The first pathway depends on a gene-for-gene interaction [32]. In this mode, the receptor protein of tomato directly interacts with the nematode effectors [33]. According to Flor’s theory, the inheritance of both resistances in the tomato and the RKN’s ability to cause disease are controlled by pairs of matching genes. The first gene, like the *mi-1* gene, is in the tomato, and the other one is in RKNs and is called a virulence (*Avr*) gene. One of the responses of this type of defense is localized programmed cell death (PCD), one of the most important responses. This is a type of hypersensitive response (HR) [33,34] (Figure 3). After the nematode enters the root of the plants; the nematode *Avr* genes produce effectors that trigger the production and the expression of plant *Mi*-resistant genes in an incompatible interaction. The result, because of this theory, is that no feeding site (giant cell) is formed.

The second defense mode is not a direct gene-for-gene interaction, but an alternative mode called the guard hypothesis. The mechanism in this theory consists of pathogen effectors that trigger the virulence factors/protein of the plant, which finally induces R-gene [35]. In these cases, the virulence factor of nematodes (*Avr* genes) interacts with tomato accessory protein, resulting in some modification of this accessory protein, which allows for the recognition by plant NBS-LRR proteins that monitor for infection. The last result of this indirect interaction is the prevention of the production and growth of nematodes by the inhibition of the formation of feeding sites.

## 5. The Genetics of Natural Resistance

### 5.1. Meloidogyne Incognita (Mi) Genes

In tomatoes, ten genes, *Mi-1* to *Mi-9* and *Mi-HT*, for resistance to root-knot nematodes have been reported [11,29]. They all originate from wild species, and only *Mi-1* has been reported as available for controlling the disease [29]. All of these genes are not stable at high temperatures, but *Mi-2*, *Mi-3*, *Mi-4*, *Mi-5*, *Mi-6*, *Mi-9*, and *Mi-HT* are heat-stable, and because of cross incompatibility, they have not been transferred successfully from wild species to cultivated tomatoes.

#### 5.1.1. Mi (Mi-1)

Plants have several defense mechanisms against a wide range of pathogens, such as R-genes. One of these genes is the *Mi-1* gene, which was first recognized in the wild tomato *S. peruvianum*, and then introduced into cultivated tomato (*L. esculentum*). *Mi* confers active resistance against several species of RKNs (*Meloidogyne* spp.) [6], especially against three RKNs (*M. arenaria*, *M. incognita*, and *M. javanica*) [36]. The *Mi-1* gene used in the management strategy of resistance against nematodes succeeds in several tomato cultivars with a high rate of resistance [37]. Until now, *Mi-1* is the only commercial source of resistance in tomato crops, but it has two significant problems: It is in active at soil temperatures above 28 °C [38], and it does not show resistance against all types of nematode-like *M. hapla*. The protein the *Mi-1* gene contains 1257 amino acids, and two of its three exons are translated. It belongs to the NBS-LRR class of R-genes that encode nucleotide-binding sites and leucine-rich repeats and includes a putative coiled-coil (CC) domain preceding the NBS [39]. *Mi-1* mapped on the short arm of chromosome 6 (Figure 4). This region of chromosome 6 in various *Solanum* species is an essential region of R-genes effective against several crop pathogens [40]. The *Mi-1* locus and the surrounding structure of chromosome 6 have been identified using linked genetic markers [41,42,43].

*Mi-1* and its homologs are grouped into two clusters with three and four copies separated by 300 kb. In 1998, three genes, *Mi-1.1*, *Mi-1.2*, and *Mi-1.3*, were identified in the *Mi* locus, but only the *Mi-1.2* gene confers resistance to RKNs [26,44]. Now there are other homologs of *Mi-1* from *Mi-1.4* to *Mi-1.7*. These seven homologs are grouped in two clusters, P1 and P2, are separated by 300 kb, and exist within a 650 kb region introgressed from *S. peruvianum*. Other studies indicate that *Mi-1.1*, *Mi-1.2*, and *Mi-1.3* are the strongest in conferring resistance against *Meloidogynes* pp. Located on the cluster P1 [40], but *Mi-1.2* from all *Mi* genes is highly specialized in resistance [45].

#### 5.1.2. *Mi-2*

The *Mi-2* gene is a single dominant gene expressed at 30 °C as non-allelic to *Mi-1*. This gene is also considered as monogenic in resistance against RKNs. The *Mi-2* gene shows heat-stable resistance to *M. incognita* in some tomato accessions such as P.I.270435-2R2 and has not been mapped [8]. Some accessions of *L. peruvianum* (P.I.270435 and P.I.126443) were resistant not only to *M. incognita*, *M. arenaria*, and *M. javanica* but also to *M. hapla*. These accessions not only were resistant at normal temperature but also confer redresistance when the soil temperature was 30 and 32 °C [46,47]. Roberts et al. [48] observed the F1 resulting from the mating between two of these wild tomato lines. *L. peruvianum* (L.p.) P.I.270435 and L.p.var. *glandulosum* 126443 were resistant to virulent selected populations of *M. incognita*, which is able to reproduce in tomato plants bearing the *Mi* gene. More knowledge about the genetic basis of this heat-stable resistance (HSR) to RKN sin *L. peruvianum* will lead to the incorporation of genes that control resistance in this modified tomato cultivars and fresh market tomato cultivars.

#### 5.1.3. *Mi-3*

*Mi-3* is a dominant gene that confers resistance against nematode strains at 32 °C, a temperature at which *Mi-1* is not active [10,20]. *Mi-3* is mapped to the telomeric region of the short arm of chromosome 12 in tomatoes [20,49] (Figure 4). There have been many attempts to transport stable heat resistance via traditional breeding, but there has been no success [50,51]. Another strategy is to clone the *Mi-3* gene from *S. peruvianum* and transfer the gene to cultivated tomatoes by plant transformation. These strategies will be possible after learning the genetic position of *Mi-3*. *Mi-3* confers resistance to *Mi-1*-virulent nematodes at 27 °C and *Mi-1*-avirulent nematode strains at 32 °C [12]. However, the two phenotypes were controlled by linked genes, *Mi-3* and *Mi-5*, as suggested by Veremis and Roberts [10]. The difference between the two proposalsis that Veremis and Roberts [10] used abridge line, EPP-1, a complex hybrid of *S. peruvianum* and *S. lycopersicum*, as their susceptible parent.

Yaghoobi et al. [12] suggested that the resistance of nematodes is more effective in homozygous plants for *Mi-3* than in heterozygotes, though both homozygotes and heterozygotes display strong resistance. *Mi-3* has been mapped to chromosome 12, but there are other disease-resistant genes that have been assigned to chromosome 12 of solanaceous plants, including resistance genes against cyst nematodes in potatoes [52], cucumber mosaic virus (CMV), a virus resistance gene in tomatoes [53], and *Me3* and *Me4*, resistance genes against RKNs in peppers [54]. However, few common markers have been used to map the R-genes on chromosome 12, and no more information to confirm whether *Mi-3* is allelic to any of these other genes is available [55]. The same phenomena of resistance to the nematodes under high temperatures are in the *Me3* gene of pepper-like *Mi-3* in tomatoes [54].

#### 5.1.4. *Mi-4*

The *Mi-4* gene is proposed to confer resistance at high temperatures against root-knot nematodes (RKNs). There have been few studies done about this gene and its mapping [10].This gene confers stable resistance under high temperatures; this has been confirmed by Veremis and Roberts [10], who found that the clone LA1708-I was resistant to *Mi*-avirulent *M. incognita* at high temperatures but was susceptible to *Mi*-virulent *M. incognita* isolates. There are two theories about the source of heat-stable resistance genes against RKNs in both *L.peruvianum* “Maranonrace” accessions LA1708 and LA2172; the former is thought to be derived from the same source; they have the same genes, but the other ones are different. The evidence that supports this is mentioned by Veremis [56], who found that different genes confer resistance at high temperatures in clones 3 MH and 2R2 of *L. peruvianum* accession PI270435. The other evidence was found in experiments with clones of *L. peruvianum* accession PI129152. The resistance in the LA1708 genotype at 32°C to *M. arenaria* isolates indicates that it occurred because of the independent resistance gene (symbol *Mi-4*) [10].

#### 5.1.5. *Mi-5*

*Mi-5* is a heat-stable resistance gene. It is located in the telomeric region of chromosome 12 in the same position of *Mi-3* and is linked together and expressed as a dominant gene (Figure 4) [10,20,57,58]. Both *Mi-5* and *Mi-3* are found in the PI126443 clone 1MH, and it is suggested in the studies of Veremis and Roberts [10] that both heat-stable resistance genes probably operate as a single locus under most conditions involving weak linking.

#### 5.1.6. *Mi-6*

*Mi-6* is a dominant gene that confers resistance against root-knot nematodes under high temperatures. It was found in the *L. Peruvianum* PI270435 clone 3MH. There are weak links between *Mi-6* and *Mi-7*, although the location of this gene on the chromosome is not known. There is also independence between the gene *Mi-6* in clone PI270435-3MHand *Mi-5* in 126443-1MH [10].

#### 5.1.7. *Mi-7*

*Mi-7* in clone PI270435-3MH confers resistance against root-knot nematodes at moderate (25 °C) temperatures. Both *Mi-7* and *Mi-6* are weakly linked and are expressed as single dominant genes [10]. There is independence between *Mi-7* and *Mi-8* genes, and they are each distinct from *Mi-3* [20]. *Mi-7,* like gene *Mi-6* and *Mi-2,* is not mapped.

#### 5.1.8. *Mi-8*

*Mi-8* is a dominant gene that was found in clone PI270435-2R2. It confers resistance against the RKN-like *Mi-7* at moderate (25 °C) temperatures. There are weak links found between *Mi-8* and *Mi-2*. *Mi-8* is distinct from *Mi-3* [10], although it has not yet been mapped. *Mi-8* seems to have a similar resistance mechanism as *Mi-2* [49]. The resistance mediated by *Mi-8* to the *Mi*-virulent *M. incognita* is accompanied by a hypersensitive response (HR). Nonetheless, some studies indicate that *Mi-8*-mediated resistance may be similar to *Mi*-mediated resistance, where cell death has been noticed near the head of the feeding J2. Recent studies have shown that cell death could be the cause of the resistance mediated by *Mi* [58].

#### 5.1.9. *Mi-9*

*Mi-9* shows resistance to all three common types of the root nodes in warm climates. Even though *M. arenaria*, *M. incognita*, and *M. javanica* are slightly different in resistant plants, all reaction shave been classified as resistant. *Mi-9* is considered as a dominant gene that shows resistance against RKNs in tomatoes at high temperatures. Previous work also has shown that *Mi-9* is not effective against *M. hapla* and the vital types of *Meloidogyne* species that can perceive tomatoes carrying a *Mi-1* gene [13,14,59]. *Mi-9* has the same effect as on the expression that *Mi-1* does in terms of *Meloidogyne* specificity, but the only difference in the phenotypic effect is the stable resistance at high temperatures. RNA silencing experiments confirmed that *Mi-9* is considered a homolog of *Mi-1*. The *L. peruvianum* accession LA2157 is highly resistant to *Mi-1*-avirulent root-knot nematodes at 25 and 32 °C but showed no resistance to *Mi-1*-virulent nematodes [47,59,60,61]. *Mi-9* is mapped on the short arm of chromosome 6 between two markers (C32.1 and C8B) (Figure 4). Six markers were used to map *Mi-9*; two of them are based on RFLP (C32.1 and C264.2), and four are based on PCR (CT119, REX-1, APS-1, and C&B) [13,14].

#### 5.1.10. *Mi-HT*

*Mi-HT* is a dominant gene that confers resistance agains troot-knot nematodes at high temperatures (32 °C). *Mi-HT* is mapped at the short arm of chromosome 6 close to the positions of *Mi-1* and *Mi-9* (Figure 4). Four markers in two clusters were adapted to map *Mi-HT*. Three markers—Mi, REX-1, and SSR-W415—formed acluster, and the resistance co-segregating with marker W737 formed another cluster [25]. The *Mi-HT* gene is different from *Mi-9*, which is due to two markers C&B and DO, and W737 could distinguish between ZN17 and the source of *Mi-9* (LA2157). Therefore, *Mi-HT* is considered as a new resistance gene found in the tomato source ZN17.

### 5.2. Elements Mediating R-genes

Several common components, elements, or genes required for *R* function or interaction with R proteins have been recently identified [62]. The function of these elements has been achieved by using mutational analysis and other molecular methods such as virus-induced gene silencing (VIGS). One of these components is a glycosyl-transferase, which was upregulated during *M. incognita* infection [63]. The other one is mitogen-activated protein kinase (MAPK), which play sasignificant role in the *Mi-1*-mediated resistance response. The silencing of three different MAPKs indicates arole for at least one MAPK cascade operating downstream of *Mi-1* [64]. Another gene that enters R-gene-mediated resistance is the transcribed product (*Sgt1*) that interacts with *Rar1* [65]. Moreover, a molecular chaperone, heat-shock protein *Hsp90*, interacts with *Rar1* and *Sgt 1* in the resistance response of the R-gene [66]. The roles of both *Sgt1* and *Hsp90*-*1* as mediated resistance provideus with more evidence for common components in early resistance gene defense signaling during pathogen infection.

Furthermore, *Rme1* plays arole in the early stages of infection and might interact directly or indirectly with *Mi-1*. Many studies indicate that the role of *Rme1* is limited to *Mi-1*-mediated resistance only and plays no role in disease resistance [9]. Another compound is nicotinamide adenine dinucleotide (NAD), which induces resistance against RKN pathogenicity, likely through the accumulation of tomato basal defense responses rather than the direct effect on the infective juvenile behavior [67].In addition, the role of *SlWRKY3* acts as a positive regulator for the resistance against the RKN-like *M. javanica* by activating lipids and the hormone-mediated defense-signaling pathway. The overexpression of *SlWRKY3* decreases infection, while its silencing results in increased infection [68]. Salicylic acid (SA) also plays a downstream role during nematode infection [69]. More studies on the elements of genes or components that mediate resistance plant response will improve the understanding about resistance against RKNs.

## 6. Problems of Natural Resistance

### 6.1. Not Effective against All Types of Nematode

The *Mi* gene is effective against the three major tropical and sub-tropical RKNs (*M. arenaria*, *M. incognita,* and *M. javanica*), but it does not work against another tropical RKN, *M. enterolobii* [70]. *Mi* is also ineffective against *M. hapla*, the northern root-knot nematode, which is common in the Northern United States and Canada. *M. enterolobii* can break the resistance of the *Mi-1* gene in tomatoes and peppers; this makes it difficult to control this type of root-knot nematode, especially in organic farming systems where chemical control is not preferred [71]. The same problem has been recorded with Florida isolates of *M. mayaguensis,* which overcome the resistance of tomato and pepper (*Capsicum annuum* L.) genotypes that contain three genes, namely the *Mi-1*,*N*, and *Tabasco* genes [70]. These are all sources of resistance in tomatoes and peppers against the three most distributed root-knot nematode species: *M. incognita*, *M. javanica*, and *M. arenaria* [71,72,73]. Furthermore, many studies have recorded a broader host range and increased pathogenicity of *M. mayaguensis* compared to other *Meloidogyne* spp. [74,75].

### 6.2. Varieties of Resistance Breakdown

Nematode-resistant varieties with the *Mi*-gene are the first defense line against RKNs. However, it has been noticed that lesions become insensitive to the *Mi* gene, the reasons for which are unknown. This is more likely when lesions are continuously exposed to root-knot resistant cultivars, especially in monoculture systems. Resistance-breaking populations of *M. incognita* have been reported since the 1990s, and these have become wides pread in some tomatoes growing in Californiaa reas [76], similar to other countries such as Greece, Spain, France, Cyprus, Italy, and Morocco [77,78,79,80,81,82,83]. The reasons for this phenomenon are not clear. It may be due to environmental factors such as temperature, changes in nematodes, and changes in resistant plants. One problem is that all resistant tomato varieties have the same origin of resistance, namely the *Mi* gene, as a result of a single hybridization between the wild tomato plant and the commercial tomato plant that was manufactured in the early 1940s. There is no significant difference between more than one resistance line because they all have the same resistance gene source. Although resistant varieties still help fight nematode infection, researchers prefer the direction of enhancing the effectiveness of resistant strains. Additionally, reducing root-knot nemato desusing chemical pesticidesor croprotation can help reduce pressure on resistance lines. Finally, there is an urgent need for plant breeders and researchers to introduce other types of resistance genes that do not depend on *Mi*-genes.

### 6.3. Genetics of Virulence in Nematodes

In most nematode species, the production of an embryo from a female gamete without any genetic contribution from a male gamete, with or without the eventual development into an adult, is called parthenogenesis.In this mechanism, there is no need for a meiotic process. Few RKN species, including *M. hapla*, are bred by facultative meiotic parthenogenesis [29]. According to this, there is significant volatility within and between species for their host range as well as virulence/avirulence motion, albeit decreases in parthenogenic RKNs pecies [82,84,85]. After repeated cultivation of resistant tomatoes under both field and greenhouse conditions, virulence populations have been previously detected from avirulent strains [80,86,87].

### 6.4. The Temperature Factor

Many effectors such as heat, soil moisture, and host variety, play major roles in the life cycle of nematodes [88]. Temperature is a vital effector because it affects both tomato resistance and the metabolic and developmental rates of nematodes. The major problem is that the resistance breaks at high soil temperatures (>28 °C) [38,47]. The temperature plays dual roles in both the resistance gene and the life cycle of the nematode.The nematode becomes active when the soil temperature is 18–32 °C. The temperature factor is the essential system to control nematodes because it only works if the soil temperatures are below 27 °C. Regarding plants, increased gall formation has been shown in plants exposed to soil temperatures a bove 28 °C (Figure 5). There is a correlation between the increase in both heat and gall numbers [24,38,47,89]. However, there is some contradiction among previous studies. Most articles have reported acomplete loss of resistance at high soil temperatures (≥32 °C) [38,90].

Others have shown that there is active resistance in some cultivars at soil temperatures ≥ 34 °C [86,91]. In some tomato plants, the resistance against nematodes is still active at high soil temperatures; this is due to the existence of one or more sources of heat-stable resistance mentioned above.

## 7. Different Approaches to Strengthening Natural Resistance

### 7.1. Marker-Assisted Selectionin Breeding Programs

Marker-assisted selection (MAS) means the use of a binding pattern of linked molecular (DNA) markers for indirect selection in the desired plant phenotype. MAS is based on the concept that the presence of a marker that is tightly linked to the gene of interest indicates the presence of that gene.The improvement of new resistance plants has many benefits. The two most important benefits of using molecular breeding are first that it is less harmful to the environment than pesticides, and second that it is less expensive. Tomatoes are considered one of the most optimal plants for using molecular markers in commercial breeding [92]. Moreover, molecular markers linked to the *Mi-1* gene have enabled the rapid screening of resistance alleles, without requiring nematode inoculation. The use of molecular marker technologies in sync with new breeding techniques is promising for the a dvancement of tomato breeding. Molecular markers used in resistance breeding against RKNs are summarized in Table 2.

### 7.2. Genetic Engineering in Controlling RKN

Although molecular breeding is the method that is most applied to achieve resistance against root-knot nematodes in tomato plants, genetic engineering is a future aspiration for further increases in resistance.

#### 7.2.1. Transfer Resistance Genes

This strategy is based on two foundations. The first is the transfer of a resistance gene from other plants to tomatoes. The second is the transfer of the *Mi* resistance gene from resistant varieties to susceptible one with high production qualities. Several resistance genes from different plants have been successfully transferred to tomatoes. These tomatoes transformed with new genes reduce diseases in transformed plants [97,98]. Some dominant genes from some crops such as the plum Myrobalan Carrie *Ma*gene [99] and taro *Colocasia esculenta* carried a cysteine proteinase inhibitor [100]. Transgenic tomatoes with these genes would be novel sources for resistance against root-knot nematodes. Moreover, cloned *Mi-1* is a good candidate for transfer to susceptible plants [26]. There are more difficulties in understanding the mechanism of R-genes in other plants of the sames pecies or plants of another family. There have been many contradictions in previous studies in the case of other transformed solanaceous plants with the *Mi-1* gene. For example, in tobacco, none of the 19 transformed lines acquired resistance against RKNs [90], but resistance has been found in eggplant [101]. The same failure has been found in more distant families, such as Arabidopsis and lettuce [101]. Transgenic tomato plants with *CeCPI*+ *PjCHI*-1 genes from Taro and *Paecilomyces javanicus* fungus showed reduced chitin content and retardation in embryogenesis in nematode eggs [102]. Additionally, transgenic tomatoes with the *Cry6A* gene from *Bacillus thuringiencis (Bt)* reduced the reproduction rate of *M. incognita* [103].

#### 7.2.2. Resistance Effectors

Proteinase inhibitors (PIs) are one of the most promising methods for managing nematodes. Proteinase inhibitors are protein molecules secreted by pathogens, which inhibit the function of proteinases [104]. This proteinase, in some cases, as a modified rice cystatin gene (a cysteine proteinase inhibitor) in transgenic Arabidopsis, blocks digestive processes in nematode feeding, resulting in, by reducing the size of the female of *M. incognita*, partial resistance [105]. In cases where cystatin is directed with a promoter that is preferentially active in the root, in potato plants, the transgenic expression is limited [106]. Different types of proteinase have been identified in tomatoes. The cathepsin L-like cysteine proteinase *Mi-cpll* [107] and serine proteinase *Mi-ser1* [108]. The anti-nematode potential of the plant was first found in transgenic potatoes expressing the serine, the cowpea (*Vigna unguiculata*) trypsin inhibitor (CpTI) against PCN (*Globodera pallida*) [109]. Oc-I1D86 was effective against different nematode species in various plants pecies [106,110,111,112,113,114]. Similarly, the overexpression of cystatin Oc-I1D86 in the Arabidopsis plants suppresses both the growth and fecundity in *M. incognita* and *H. schachtii* [109].

#### 7.2.3. Gene Silencing

In recent years, RNA interference (RNAi) has become a powerful approach for developing nematode resistance. RNAi can be used to reduce levels of chitin synthase transcripts, either by feeding *C. elegans* with suitably transformed *E. coli* or by soaking *M. artiellia* eggmasses in dsRNA [115]. Yadav et al. [116] used RNAi technology in reducing galling and female numbers. Additionally, Huang et al. [117] used RNAi insub-ventral pharyngeal glands for reducing galling and a number of established nematodes. Targeting Mi-cpl-1 by RNAi resulted in a reduction in the abundance of the corresponding transcript and a reduction in cysteine proteinase activity in J2*M. Incognita* [118]. Choudhary et al. [119] knocked down AF531170,a parasitism gene, usingRNAi, reducing the number of developing nematode females 7, 21, and 30 days post-infection.

Niu et al. [120] used RNAi to knock down the Rpn7 gene of *M. incognita*, resulting in a reduction of nematode infection ranging from 55.2 to 66.5%, and the a mount of eggmass pergram root tissue was reduced by 34%. Transgenic expression of the RNAi construct of the *Mi-cpl-1* gene resulted in a 60–80% reduction in the infection and multiplication of *M. incognita* in tomatoes [121].

The newest approach to confer resistance in the tomato planta gainst RKNs is the CRISPR-Cas9 strategy because tomato transformation takes a long time to modulate resistance. Researchers at California University began a new project on 20 February 2019, under the title “Variability, Adaptation, and Management of Nematodes Impacting Crop Productionand Trade”. This project is aimed at genetic characterization and biological variation in nematodes relevant to crop production and trade. To achieve this goal, they will establish a hairy root system using the CRISPR-Cas9 to develop resistance against RKN. The development of hairy root transformation will be used to evaluate CRISPR-Cas9 vectors that cause negative regulators of plant immunity [122]. More knowledge and progress in the genetics and inheritance of both tomatoes and nematodes will increase our understanding of plant–nematode interactions. Additionally, further identifying both host and pathogen genes involved in the infection stage will provide us with more tools for controlling the resistance against RKNs in tomatoes.

## Figures and Tables

**Figure 1 genes-10-00925-f001:**
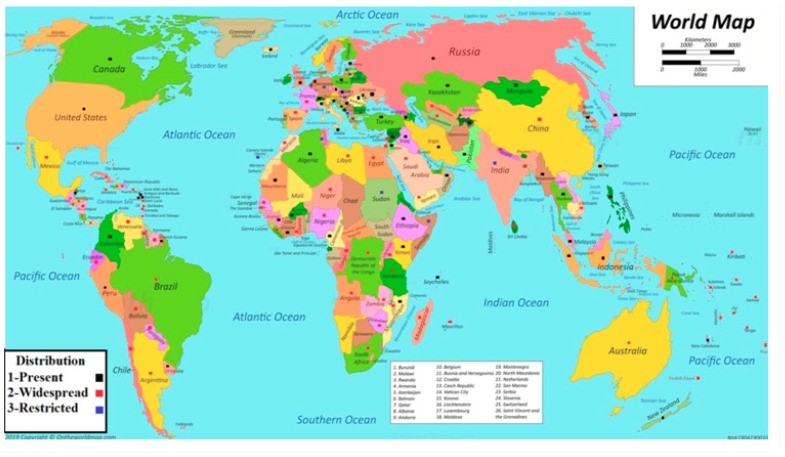
Distribution of *Meloidogyne* species; three types of distribution: (1) black marks indicate present, (2) red marks indicate widespread, and (3) blue marks indicate restricted.

**Figure 2 genes-10-00925-f002:**
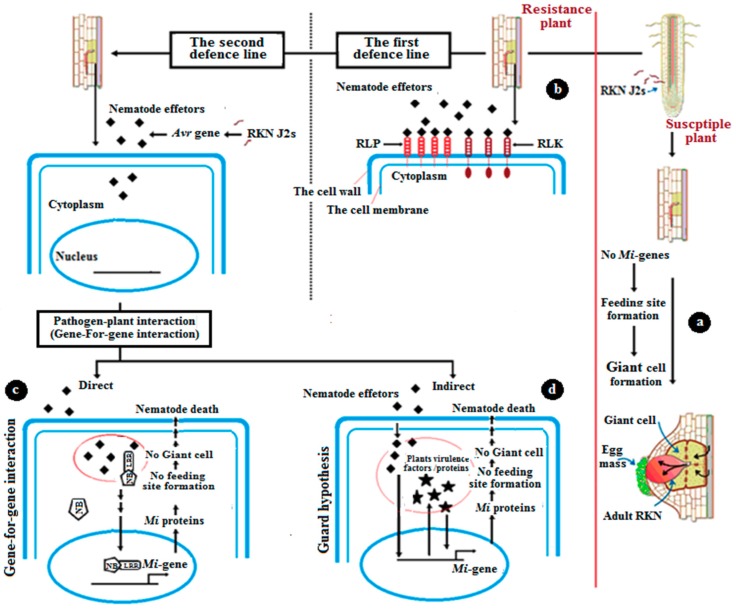
The mechanism of natural resistance against the root-knot nematode (RKN). (**a**) In susceptible plants, where there are no *Mi*-genes, the nematode completes its life cycle in the root by forming giant feeding cells. (**b**) In the resistance case, the plant undergoes the first defense line against RKN penetration by the interaction between extracellular receptor proteins, receptor-like kinases (RLK), receptor-like protein (RLP), and nematode effectors. (**c**) The plant then begins the second defense line, which includes direct gene-for-gene interaction. This theory depends on direct interaction between the receptor protein of tomatoes and nematode effectors, producing *Mi*-proteins, which prevent the nematode from feeding. No giant cell formation is observed. (**d**) The other second defense line is an indirect pathway, which is referred to as the guard hypothesis. In these cases, the virulence factor of the nematode (*Avr* genes) interacts with tomato accessory protein.

**Figure 3 genes-10-00925-f003:**
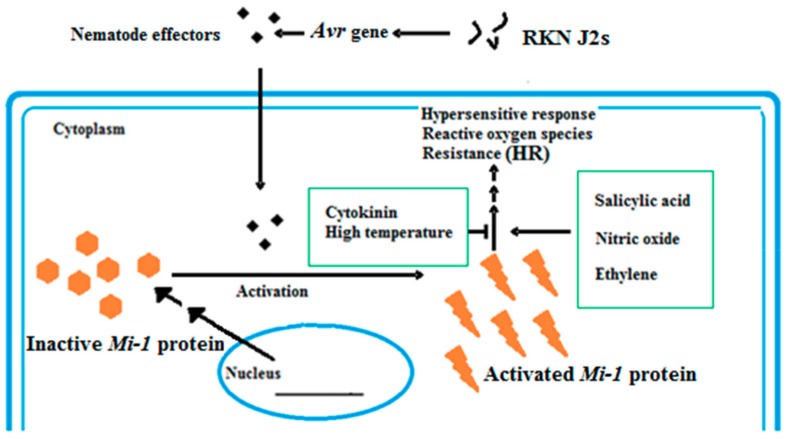
Hypersensitive response of *Mi-1* after nematode infection. The nematode *Avr* genes trigger the tomato *Mi-1* resistance gene(R-gene) to be active under the salicylic acid pathway with inhibition by both cytokinin and high temperature.

**Figure 4 genes-10-00925-f004:**
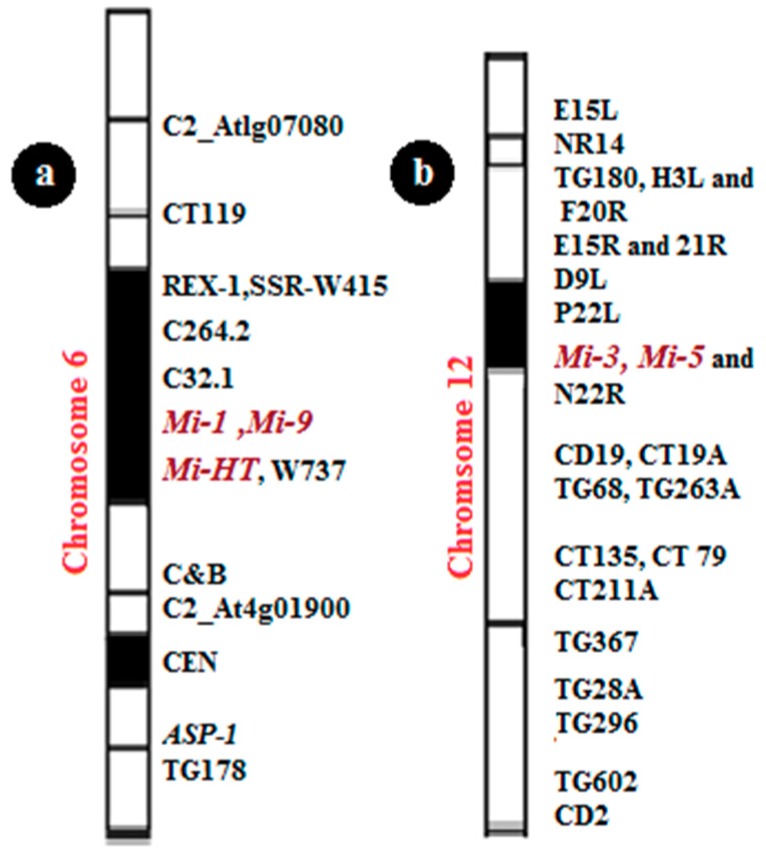
Mapping of *Mi* genes on tomato chromosomes; (**a**) the site of *Mi-1*, *Mi-9*, and *Mi-HT* on chromosome 6; (**b**) the site of *Mi-3* and *Mi-5* on chromosome 12.

**Figure 5 genes-10-00925-f005:**
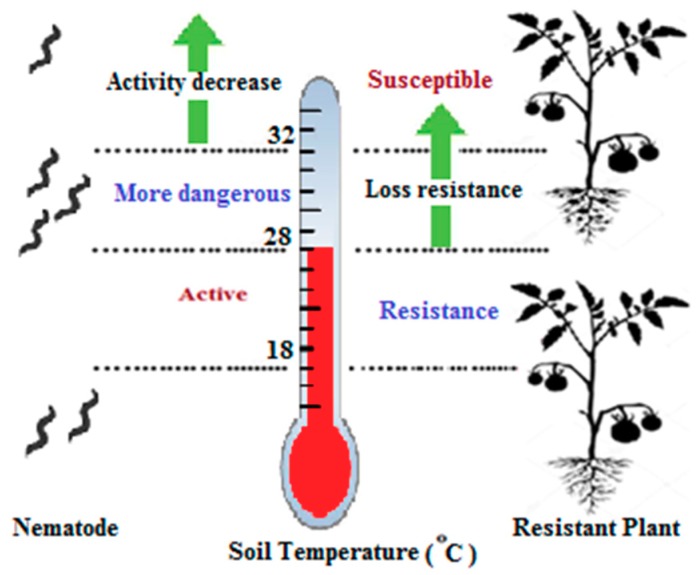
The effect of temperature on both resistant plants and nematodes. A temperature from 18 to 28 °C is suitable for both plants and nematodes. With a temperature up to 28 °C, the *Mi-1* gene in resistant plants becomes inactive, and at the same time, the nematode becomes more dangerous. The nematode activity decreases when the temperature exceeds 32 °C.

**Table 1 genes-10-00925-t001:** Tomato plants with *Mi*-genes.

Genotype/Lines/Cultivars	*Mi* Genes	Notes	Reference
(*L. peruvianum*) PI128657	*Mi-1*	High-level resistance and the main source of resistance.	[6]
(*L. esculentum*) VFNT	*Mi-1*	Resistance	[20,21]
(*L. esculentum*) Mobile	*Mi-1*	Resistance	[21]
(*L. esculentum*) Ontario	*Mi-1*	Resistance	[21]
*Solanum lycopersicum* cv. Amelia	*Mi-1*	Resistance	[22]
(*L. esculentum*) CLN2026C	*Mi-2*	Resistance	[23]
(*L. esculentum*) CLN2026E	*Mi-2*	Resistance	[23]
(*L. esculentum*) CLN1464A	*Mi-2*	Resistance	[23]
*(L. peruvianum)* 2R2-clone PI270435	*Mi-2*	Heat-stable resistance	[8,24]
(*L. esculentum*) VWP2	*Mi-3*	Heat-stable resistance	[12]
*(L. peruvianum)* 1MH-clone PI126443	*Mi-3*	Heat-stable resistance	[24]
(*L.peruvianum*) *Maranon* LA1708	*Mi-4*	Heat-stable resistance	[10].
(*L. peruvianum*) 1MH-clone PI126443	*Mi-5*	Heat-stable resistance	[10]
*(L.peruvianum)* 3MH-clone PI270435	*Mi-6*	Heat-stable resistance	[10]
(*L.peruvianum*) 3MH-clonePI270435	*Mi-7*	Resistance to RKN, including strains virulent on *Mi*	[10]
*(L. peruvianum)* 2R2-clone PI270435	*Mi-8*	Resistance to RKN, including strains virulent on *Mi*	[10]
(*S.Arcanum*)\LA2157	*Mi-9*	Heat-stable resistance	[13,14]
ZN48	*Mi-HT*	Heat-stable resistance	[25]
ZN17	*Mi-HT*	Heat-stable resistance	[25]
LA0385		Heat-stable resistance	[25]
CastlerockII		Resistance	[26]
Sun6082		Resistance	[26]
(*S. lycopersicum*) Tomato Mongal T-11		High resistance	[27]
(*S. lycopersicum* L) Samrudhi F1		Resistance	[28]
(*L.esculentum*) LE812		Resistance	[23]

**Table 2 genes-10-00925-t002:** Molecular markers related to the root-knot nematode resistance.

No.	MarkerName	Marker	Gene	Sequence	Reference
**1**	C&B	CAPS	*Mi-9*	5′-TACCCACGCCCCATCAATG-3′ 5′-TGCAAGAGGGTGAATATTGAGTGC-3′	[14]
**2**	Mint-1	SCAR	*Mi-1.1*, *1.2*, *1.4* and *1.6*	5′-TTCTCTAGCTAAACTTCAGCC-3′ 5′-TTTTCGTTTTTCCATGATTCTAC-3′	[13]
**3**	TG180	SCAR	*Mi-3*	5′-ATACTTCTTTRCAGGAACAGCTCA-3′ 5′-ACTTAGTGATCATAAAGTACCA-3′	[12]
**4**	REX-1	CAPS	*Mi-1.2*	5′-TCGGAGCCTTGGTCTGAATT-3′ 5′-GCCAGAGATGATTCGTGAGA-3′	[14,93]
**5**	JB-1	CAPS	*Mi-1*	5′-AACCATTATCCGGTTCACTC-3′ 5′-TTTCCATTCCTTGTTTCTCTG-3′	[94]
**6**	PMi12	SCAR	*Mi-1*	5′-CCTGCTCGTTTACCATTACTTTTCCAACC-3′ 5′-CTGCTCGTTTACCATTACTTTTCCAACC-3′	[95]
**7**	Mi23	SCAR	*Mi-1.2*	5′-TGGAAAAATGTTGAATTTCTTTTG-3′ 5′-GCATACTATATGGCTTTTTACCC-3′	[40]
**8**	*APS-1*	CAPS	*Mi*	5′-GGATTTTCGTGTTCTTGGTG-3′ 5′-GCCCAGTCAGCAAGAAAACT-3′	[14]
**9**	CT119	CAPS	*Mi*	5′-TCAGGTATCGAACCAAAACC-3′ 5′-TAAAAGGTTCATCCTAATAC-3′	[14]
**10**	C1/2	RAPD	*Mi1.1*	5′-CAGTGAAGTGGAAGTGATGA-3′	[20,96]
**11**	C2S4	RAPD	*Mi1.2*	5′-CTAAGAGGAATCTCATCACAGG-3′	[20,96]
**12**	TG-263	SCAR	*Mi-3*	5′-GCTGAGAAATAAAGCTCTTGAGG-3′ 5′-TACCCTTAATGCTTCGGCAGTGG-3′	[12]
**13**	*Cf-2*	CAPS	*Mi-1.1*, *1.2* and *1.3*	5′-CTAGGCAGCGATTTCCATTT-3′ 5′-CGGAATAGGTAATGGCCTTC-3′	[13]

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
