# Peer review of "Tomato Natural Resistance Genes in Controlling the Root-Knot Nematode"

_genes, 2019, doi:10.3390/genes10110925_

Round 1

Reviewer 1 Report

The article points out an interesting field,  however the  review has some significant pitfalls.

In general, many key concepts are not explained in detail as expected by an interested audience. The figure captions have to be self explicative, moreover the resolution of figures needs to be improved.

English style also needs a major revision, revising the grammar and punctuation (see the pdf file with some suggestions, as a major revision is needed only some corrections are shown)

a brief description of plan parasitic nematodes biology and damage should be added in the introduction.

In the references it is better to consider research article, not other reviews

Author Response

Thank you very much for these supportive comments; we highly appreciate your great effort and valuable time you spent in reviewing our paper. We respond to all the suggested corrections and we wish these changes would be as you expected from us.

Comment 1: Many key concepts are not explained in details as expected by an interested audience. The figure captions have to be self explicative, moreover the resolution of figures needs to be improved.

Response: I've edited the images to make it clearer and I've added comments underneath the images that I hope will be appropriate to explain and clarify the image.

Comment 2: English style also needs a major revision, revising the grammar and punctuation (see the pdf file with some suggestion, as a major revision is needed only some corrections are shown).

Response: English language was revised by MDPI Language Editing Services; many corrections were made (Tracked version and certificate are attached). Also, we have responded to all comments shown in the pdf in addition to modification which done by MDPI Language Editing Services.

The comments in the attached pdf:-

Thanks very much for this careful review and for your great time.

line 23: Second > second

Response: Done (line 24).

line 23 and 24: more cultivated? or more consumed?

Response: Done (line 24,25). Added this “the second in global food production.”

line 26: neuro > neurological

Response: Done (line 27).

line 28: Deleted what is mention, and replace by (The root-knot nematode (RKN) belongs to)

Response: Done (line 30).

line 29: species > of them.

Response: Done (line 31).

line 29: Meloidogyne

Response: Done (line 31).

line 30: consider > are considered

Response: We corrected it (lines32).

line 31: This genus is > They are

Response: Done (line 32).

line 31: which, the host is > capable of

Response: Done (line 33).

line 33: motivate > induce

Response: Done (line 35).

line 33: the giant cells > giant feeding cells

Response: Done (line 35).

line 33 to 36: all line from 33 to 36 > leading to a decrease in plant nutrition and water uptake. As a consequence, plants can show several symptoms such as wilting and stunting, thus increasing the susceptibility to other pathogens and with considerably reduced yields.

Response: We deleted this sentence and replaced by the mentioned one but with small change according to the native speaker (line 35 to 38).

line 38: root-knot nematodes > RKN

Response: Done (line 56).

Line 38: was > was first.

Response: Done (line 56).

line 41: This gene confers resistance to > revealed important against

Response: We deleted it and replaced by revealed important against but with small change according to the native speaker (line 59 to 60).

Line 42 to 44: “Mi-1 is the first major gene” to “M. javanica. It” line 44 > Mi-1

Response: We deleted the mentioned then replaced by Mi-1 (line 60).

Line 44 and 45: The Mi gene gives the resistance against the root-knot nematode to the tomato plants early after two weeks of germination > the resistance provided by the Mi gene is induced at an early stage in tomato plants, as soon as two weeks after germination.

Response: replaced but with small change according to the native speaker (line 61 and 62).

line 194 and 195: refer > refers

Response: Done (lines199 and 200).

line 97: kinas > kinases

Response: done ( line 128).

line 98: is > are

Response: Done (line 129).

line 102: mod > mode

Response: Done (line 133).

line 107: enter > enters

Response: Done (line 139).

line 108: trigged > trigger

Response: done (line 140).

line 109: The result > The result,

Response: Done (line 141).

Figure 2: gaint > giant

Response: Done to all of this word in the figure 2

line 152: root-knot nematode > RKNs

Response: Done (line 214).

line 215: CMV> cucumber mosaic virus (CMV)

Response: Done (line 281).

line 372: Myrobalam > Myrobalan

Response: Done (line 471).

References 11, 12,14,15,16,17 and 18 : there is not a real comment of this article in the text

Response: 

(11) Jablonska B, Ammiraju J. S, Bhattarai K. K, Mantelin S, Martinez de Ilarduya O, Roberts P. A, and I.Kaloshian. The Mi-9 gene from Solanum arcanum conferring heat-stable resistance to root-knot nematodes is a homolog of Mi-1. Plant Physiol.2007, 143: 1044-1054. (12) Ammiraju J. S. S, Veremis J. C, Huang X, Roberts P. A, Kaloshian , The heat-stable, root-knot nematode-resistance gene Mi-9 from Lycopersicon peruvianum is localized on the short arm of chromosome 6. Theor Appl Genet2003, 106:478–484

These found in the text in many sites like, in the text under title “5.1.9. Mi-9” and also in the two tables, first the table no.1 under title “Tomato plants with mi-genes” and the other one is the table no.2 under title “Molecular markers related the RKN resistance”.there were some erorr in putting the right references in its site but I solved it.

(14) Jones JT, Haegemen A, Danchin EGJ, Gaur HS, Helder J, Jones MGK, Kikuchi T, Palomares-Rius JE, Wesemael WML, Perry RN .Top 10 plant-parasitic nematodes in molecular plant pathology. Mol Plant Pathol.2013. 14: 946-961.

Really , this was not mentioned enough in the text so i added some of the sutible results of this paper from line 84 to 91.

(15) Abad P, Favery B, Rosso MN, Castagnone-Sereno P.Root-knot nematode parasitism and host response: Molecular basis of a sophisticated interaction. Mol Plant Pathology.2003. 4: 217–224.

It mentioned in many sites :first at lines 75 and 76 and second in the lines 205 and 206.

(16) Trudgill DL, Blok VC. Apomictic, polyphagous root knot nematode : exceptionally successful and damaging biotrophic root pathogens. Annu Rev Phytopathol 2001. 39:53- 77.

It found in the lines 75,76,79 and 80.

(17) Olsen M.W. Root-knot Nematode. University of Arizona, Arizona Cooperative Extension 2000, AZ1187 (November), 1–3

It founds in lines from 76 to 79 at the text.

(18) Ralmi N H A A, Khandaker M. M and Nashriyah M, Occurrence and control of root knot nematode in crops: A review. AJCS 2016, 10(12):1649-1654.

It founds in lines from 78 to 81 at the text.

Comment 3: A brief description of plant parasitic nematodes biology and damage should be added to the introduction.

Response: It was miss important points so, we added two paragraph; first one talking about nematode behavior during the infection (from lines 39 to 48) and the second one several damages done by it (from lines 50 to 55).

Comment 4: In the references it better to consider research article, not other reviews.

Response: It’s right but, there were some points that were mentioned only in a few researches, so a few review researches have been referenced, for example some of resistance genes (Mi-4,Mi-5,Mi-6,Mi-7 and Mi-8) are lack in papers .

Reviewer 2 Report

The manuscript (review) entitled ‘tomato natural resistance genes inn controlling root-knot nematode’ by El-Sappah et al. has assessed 1) current understanding of tomato (R)-resistant genes against RNK, 2) issues/problems within, and 3) current/future directions.

A)  I found, in particular, the great value/interest of the chapters #3, #6 and #7 to the broad audience.

B)  I largely am fine with chapter #4, except that – I believe – ‘guard hypothesis’ is a part of ‘gene-for-gene theory.

C)  I believe that it would be more informative if the chapter #5 includes the discussion of possible ID or modes of RKN-derived effectors (or lack of knowledge for those), since R-gene mediated resistance (or ETI) requires complementary effectors (avr genes) in their system.

Overall, this manuscript well summarizes interesting topics in plant-nematode interactions, and provides valuable perspectives to the audience. However, a current version requires major revision on grammar, typos, and so on throughout a manuscript. 

Author Response

Thank you very much for your observations and opinions, which directed us toward improving our paper

Comment 1: I found, in particular, the great value/interest of the chapters 3, 6 and 7.

Response: Thanks a lot, this comment from you gives me more enthusiasm to complete the scientific research process.

Comment 2: I largely am fine with chapter 4, except that-I believe-‘ guard hypothesis is a part  of  gene-for-gene-interaction.

Response: yes, that is right I modified the figure no.2 which describe this, and modified the paragraph “The second defence mode isn’t gene-for-gene interaction but alternative mode called ‘guard  hypothesis” (line 177) to “The second defence mode is not a direct gene-for-gene interaction, but an alternative mode called the guard hypothesis “

Comment 3: I believe that it would be more informative if chapter 5 includes the discussion of possible ID or modes of RKN –derived effectors (or lack of knowledge for those), since R-genes mediated resistance (or ETI) requires complementary effectors ( Avr genes ) in their system.

Response: I already added paragraph (5.2.), which contains some of elements mediate R-genes resistance (Lines from 348 to 371).

Comment 4: This manuscript well summarizes interesting topics in plant-nematode interactions, and provides valuable perspectives to the audience. However, a current version requires major revision on grammar, typos, and so on throughout a manuscript.

Response: Thanks a lot for your comment which encourage me, and about the English language, it was revised by MDPI Language Editing Services; many corrections were made (Tracked version and certificate are attached).

Reviewer 3 Report

The manuscript by El-Sappah et al reviews the natural resistance of tomato to RKN. The manuscript is well-written and covers the literature. I would just suggest editing of manuscript for language and typos. 

Author Response

Thank you very much for these supportive comments; we highly appreciate your great effort and valuable time you spent in reviewing our paper.

Comment: The manuscript by El-Sappah et al reviews the natural resistance of tomato to RKN. The manuscript is well-written and covers the literature. I would just suggest editing of manuscript for language and typos.

Response: Thank you very much Sir for your kind patience and your careful review of our paper more than once. English language was revised by MDPI Language Editing Services; many corrections were made (Tracked version and certificate are attached).

Round 2

Reviewer 1 Report

The manuscript has been improved, but some terminology is not adequate for a scientific paper.

From line 39 to 44 the information seems taken from  a report by Kenneth Seebold, you should cite it.

The bibliography has to be checked.

Please control lacking spaces between paragraphs.

Author Response

Thank you very much for these supportive comments; we highly appreciate your great effort and valuable time you spent in reviewing our paper. We respond to all the suggested corrections and we wish these changes would be as you expected from us.

Comment 1: The manuscript has been improved, but some terminology is not adequate for a scientific paper.

Response: Thanks about your comment; this improved wouldn’t have been done without your guidance, your honor. I already modified some terminology according your suggestion like:

Larva to juvenile

Comment 2: From line 39 to 44 the information seems taken from a report by Kenneth Seebold, you should cite it.

Response: I have already added it as number (3) in the text (Line 44) and in the reference as “Kenneth W. Seebold. Root-knot Nematode In Commercial & Residential Crops. Plant Pathology Fact Sheet. Plant Pathology Extension. College of Agriculture. University of Kentucky.2014, PPFS-GEN-10.” (Lines 521 and 522) 

            Comment 3: The bibliography has to be checked.

      Response: I already did this, when I added the reference no.3 so every next numbers had been changed with change the site of some references to be suitable to the text. This error happened by mistake, so I corrected it.

         Comment 4: Please control lacking spaces between paragraphs.

      Response: There already some of this mistake due to the type of office, I tried to solve it.

Round 3

Reviewer 1 Report

Please see my suggestions.

There are many spaces lacking.  I have  highlighted in yellow, in the number line and along the text, (always highlighted  between the words) the places that need to add a space.
